# Effect of Calcination Temperatures on Surface Properties of Spinel ZnAl_2_O_4_ Prepared via the Polymeric Citrate Complex Method—Catalytic Performance in Glycerolysis of Urea

**DOI:** 10.3390/nano13131901

**Published:** 2023-06-21

**Authors:** Nhiem Pham-Ngoc, Huy Nguyen-Phu, Eun Woo Shin

**Affiliations:** 1School of Chemical Engineering, University of Ulsan Daehakro 93, Nam-gu, Ulsan 44610, Republic of Korea; nhiem.phamngoc@gmail.com; 2Department of Chemical and Biomolecular Engineering, Seoul National University of Science and Technology, Seoul 01811, Republic of Korea; huycanphu@gmail.com

**Keywords:** partially inverse spinel, polymeric citrate complex method, acidity, glycerol carbonate

## Abstract

In this study, we investigated urea glycerolysis over ZnAl_2_O_4_ catalysts that were prepared by using a citrate complex method and the influence of calcination temperatures on the surface properties of the prepared catalysts by varying the calcination temperature from 550 °C to 850 °C. As the reciprocal substitution between Al^3+^ and Zn^2+^ cations led to the formation of a disordered bulk ZnAl_2_O_4_ phase, different calcination temperatures strongly influenced the surface properties of the ZnAl_2_O_4_ catalysts, including oxygen vacancy. The increase in the calcination temperature from 550 °C to 650 °C decreased the inversion parameter of the ZnAl_2_O_4_ structure (from 0.365 to 0.222 for AlO_4_ and 0.409 to 0.358 for ZnO_6_). The disordered ZnAl_2_O_4_ structure led to a decrease in the surface acidity. The ZnAl_2_O_4_-550 catalyst had a large specific surface area, along with highly disordered surface sites, which increased surface acidity, resulting in a stronger interaction of the Zn NCO complex on its surface and an improvement in catalytic performance. Fourier transform infrared and thermogravimetric analysis results of the spent catalysts demonstrated the formation of a greater amount of a solid Zn NCO complex over ZnAl_2_O_4_-550 than ZnAl_2_O_4_-650. Consequently, the ZnAl_2_O_4_-550 catalyst outperformed the ZnAl_2_O_4_-650 catalyst in terms of glycerol conversion (72%), glycerol carbonate yield (33%), and byproduct formation.

## 1. Introduction

In recent years, climate change has become increasingly severe, which has led to the promotion of the development of the biodiesel industry [1,2]. Large amounts of crude glycerol as a byproduct generated from biodiesel production have an economic impact as a cheap and commercially viable renewable feedstock [3]. Among various glycerol reaction pathways, glycerolysis into glycerol carbonate (GC) is an interesting research direction. GC is an important glycerol derivative that is considered an interesting value-added chemical due to its excellent properties such as being a non-hazardous low-vapor pressure liquid, as well as its biodegradability, its low toxicity, its non-flammability, its high boiling point, and its low volatility [4,5,6]. Because of its outstanding properties, GC has a wide range of applications, including solvents, electrolytes, surfactants, wetting agents, and as a chemical intermediate in polymer synthesis [5,7].

Traditionally, the chemical synthesis of GC is by glycerol reacting with phosgene; however, this reaction is very dangerous and not environmentally friendly [8]. Alternatively, carbonate sources with milder reaction conditions such as dialkyl carbonate [9,10,11], alkylene carbonate [12], and CO_2_ [13,14] can be used. With special advantages such as being cheap, easy to obtain, and recyclable, urea is used, along with glycerol, to efficiently produce GC [15,16,17,18]. In our previous studies, catalytic performance was affected by the existence of Zn-containing intermediates, namely, zinc diamine diisocyanate (Zn(NH_3_)_2_(NCO)_2_, abbreviated as a Zn NCO complex). The formation of the solid Zn NCO complex on the ZnAl_2_O_4_ phase offered many benefits for increasing the GC yield: acting as active sites for the heterogeneous GC pathway and interacting with an insoluble ZnAl_2_O_4_ phase [17].

The ZnAl_2_O_4_ spinel structure is represented with the typical formula (Zn^2+^)[Al^3+^_2_]O_4_, where Zn^2+^ cations occupy one-eighth of the tetrahedral sites, and Al^3+^ cations occupy half of the octahedral sites. However, a normal ZnAl_2_O_4_ spinel structure can be partially disordered, replacing some Al^3+^ cations with Zn^2+^ cations in the octahedral sites and some Zn^2+^ cations with Al^3+^ cations in the tetrahedral sites [16]. It has been reported that the disordered structure of the ZnAl_2_O_4_ spinel structure affected the formation of the Zn NCO complex. Despite many studies on the various types of reactions or methods for preparing the ZnAl_2_O_4_ spinel structure, only a few studies have been conducted to investigate the influence of calcination temperatures on the structural properties of ZnAl_2_O_4_ catalysts. The formation of the disordered ZnAl_2_O_4_ structure is highly dependent on the calcination temperature.

Hence, in this work, ZnAl_2_O_4_ spinel catalysts with a disordered structure (denoted as ZnAl_2_O_4_-X, where X represents calcination temperature in °C) were prepared by using a polymeric citrate complex method and calcined at different temperatures ranging from 550 °C to 850 °C to investigate the influence of calcination temperatures on the ZnAl_2_O_4_ spinel lattice structure, the acidity of the catalysts, and the formation of a solid Zn NCO complex on the catalysts. We also used the prepared catalysts in the glycerolysis of urea under vacuum (3 kPa) at 140 °C for 5 h. Characterizations, including Fourier transform infrared spectroscopy (FT-IR) and X-ray photoelectron spectroscopy (XPS), were employed to observe the disordered ZnAl_2_O_4_ spinel structure. Acidic sites were measured via the temperature-programmed desorption of NH_3_ (NH_3_-TPD), and the formation of a solid Zn NCO complex was demonstrated by FT-IR and thermogravimetric analysis (TGA).

## 2. Materials and Methods

### 2.1. Catalyst Preparation

Citrate complex ZnAl_2_O_4_ catalysts were prepared by using a modified citrate complex technique [19]. An aqueous solution of Zn(NO_3_)_2_.6H_2_O (Sigma-Aldrich Korea, Gyounggi, Republic of Korea) and Al(NO_3_)_3_·9H_2_O (Sigma-Aldrich Korea) with a molar ratio of Zn and Al in the precursor of 1:2 was prepared at room temperature, and the citric acid powder (citric acid to metal ratio of 2:1) (Sigma-Aldrich Korea) was added to the solution, followed by stirring with a magnetic bar. Stirring was continued at 70 °C to evaporate water until a yellow viscous gel solution was formed. The gel was further dried in an oven at 140 °C and spontaneously solidified by the emission of NO_x_ gases. Finally, all catalysts were calcined under a set temperature (550 °C–850 °C) in a furnace for 4 h. The catalysts were denoted as ZnAl_2_O_4_-X, where X indicates the calcination temperature.

### 2.2. Reaction Test

The reaction was performed in a round-bottom, three-neck 100 mL flask, in which one neck was connected to a vacuum line through a water condenser. Glycerol (Sigma-Aldrich Korea) (0.2 mol) was added to the flask at 80 °C under stirring by a magnetic bar to reduce the high viscosity of the glycerol. The reactor was connected to a vacuum pump through a HNO_3_ (Sigma-Aldrich Korea) solution trap (to remove NH_3_) and a cold trap (to remove order volatiles). After 10 min, 0.2 mol of urea (Sigma-Aldrich Korea) was poured into the flask to mix with glycerol in the solution. When all the urea was dissolved completely in glycerol and the solution was transparent, a certain amount of catalyst (5 wt% compared with the initial glycerol amount) was added to the flask. The reaction was performed under vacuum pressure (3 kPa) at 140 °C with constant stirring.

After the reaction tests, ethanol (Sigma-Aldrich Korea) was poured into the final products, and the liquid products were separated from the spent catalyst via filtration. The liquid product was quantitatively analyzed via gas chromatography by using a gas chromatography machine (Acme 6100 GC, YL Instrument Co., Ltd., Dongangu, Anyang, Republic of Korea) (a schematic diagram of the catalyst activity test device is shown in Appendix A) with a flame ionization detector and a capillary column DB-Wax (30 m × 0.25 mm × 0.25 µm). The molar amount of each component was calculated by using the internal standard method, with tetraethylene glycol (Sigma-Aldrich Korea) as the internal standard chemical. Glycerol conversion, GC selectivity, GC yield, and byproduct selectivity were calculated by using the equations below. The amount of each chemical is on the mole unit.
$$\mathrm{Glycerol}\;\mathrm{conversion}\;(\%)=\frac{\mathrm{Initial}\;\mathrm{amount}\;\mathrm{of}\;\mathrm{glycerol}\;-\;\mathrm{Residual}\;\mathrm{amount}\;\mathrm{of}\;\mathrm{glycerol}}{\mathrm{Initial}\;\mathrm{amount}\;\mathrm{of}\;\mathrm{glycerol}}\times 100\;\mathrm{GC}\;\mathrm{yield}\;(\%)=\frac{\mathrm{Amount}\mathrm{of}\mathrm{GC}}{\mathrm{Initial}\;\mathrm{amount}\;\mathrm{of}\;\mathrm{glycerol}}\times 100\;\mathrm{GC}\;\mathrm{selectivity}\;(\%)=\frac{GC\;\mathrm{yield}\;(\%)\;}{\mathrm{Glycerol}\;\mathrm{conversion}\;(\%)}\times 100\;\begin{array}{c} \mathrm{Byproduct}\;\mathrm{selectivity}\;(\%)\\ =\frac{\mathrm{Amount}\;\mathrm{of}\;\mathrm{byproduct}}{\mathrm{Initial}\;\mathrm{amount}\;\mathrm{of}\;\mathrm{glycerol}\;-\;\mathrm{Residual}\;\mathrm{amount}\;\mathrm{of}\;\mathrm{glycerol}\;}\times 100 \end{array}$$

FT-IR spectra of the liquid products were obtained by using a Thermo Scientific™ Nicolet™ iS™5 FT-IR spectrometer (Thermo Fisher Scientific, Waltham, MA, USA) by dropping a liquid sample between KBr (Sigma-Aldrich Korea) plates.

### 2.3. Catalyst Characterization

The Chemical Composition Zn/Al Molar Ratio was measured by using an Agilent Technologies 5110 ICP-OES (Agilent, Santa Clara, CA, USA) instrument. The surface characterizations were measured via N_2_ adsorption isotherm analysis on a Micromeritics ASAP 2020 (USA) apparatus. The surface area was calculated by using the Brunauer–Emmett–Teller (BET) method. X-ray diffraction (XRD) patterns for fresh catalysts were obtained by using a Rigaku RAD-3C diffractometer (Rigaku Corp., Tokyo, Japan) with Cu Ka radiation (λ = 1.5418 Å) at a scattering angle (2θ) scan rate of 2°/min, operating at 35 kV and 20 mA. The fresh and spent catalysts were analyzed by using a Thermo Scientific Nicolet iS5 FT-IR spectrometer (Thermo Fisher Scientific). XPS data were surveyed by using a Thermo Scientific K-Alpha XPS spectrometer (Thermo Fisher Scientific). The numbers of acidic and basic sites were measured on the basis of TPD-NH_3_/CO_2_ on a MicrotracBEL BELCAT-M instrument (MicrotracBEL Corp., Osaka, Japan). An amount of 100–200 mg of fresh catalysts was placed into the quartz sample tube of the instrument. Initially, the sample was pretreated under helium flow (100 mL/min) at 600 °C for 1 h and then cooled down to 50 °C. After that, a flow of NH_3_ or CO_2_ (50 mL/min) was injected for chemisorption. Finally, the temperature was increased to 600 °C with a ramping rate of 1.5 °C/min; the desorbed species were removed via helium flow (30 mL/min) and analyzed via thermal conductivity detection. The TGA of the spent catalysts was performed by using a TGA Q50 apparatus (TA Instruments, New Castle, DE, USA).

## 3. Results and Discussion

### 3.1. Characterizations of Fresh Catalysts

The textural properties of the fresh catalysts, which were calculated from N_2_ adsorption–desorption measurements, are summarized in Table 1, and the N_2_ adsorption–desorption isotherms and pore size distributions are shown in Appendix A. As indicated in Appendix A, all ZnAl_2_O_4_ catalysts exhibited type IV isotherm and type H3 hysteresis loops (based on the IUPAC classification [20]), which is typical for mesoporous materials, indicating slit-shaped pores. As the calcination temperature increased, the hysteresis loop became narrower, with a lower BET surface area and a lower average pore volume in the order of ZnAl_2_O_4_-850 < ZnAl_2_O_4_-750 < ZnAl_2_O_4_-650 < ZnAl_2_O_4_-550. In contrast, the average pore diameter exhibited an opposite trend. Increasing the calcination temperature led to the collapse of small pores to generate bigger ones. The relationships between textural properties and calcination temperatures are depicted in Figure 1.

Figure 2 depicts the XRD patterns of the fresh ZnAl_2_O_4_-X catalysts. Characteristic peaks appearing at 2θ = 31.2°, 36.8°, 44.8°, 49.1°, 55.7°, 59.3°, 65.2°, 74.1°, and 77.3° can be attributed to the crystalline phase of ZnAl_2_O_4_. All XRD patterns show ZnAl_2_O_4_ peaks, which indicates the formation of ZnAl_2_O_4_ crystallite after high-temperature calcination by using the polymeric citrate complex method. The final decomposition temperature of the precursor in the derivative TGA (DTGA) result is below 550° (Appendix A), confirming that the ZnAl_2_O_4_ crystallite structure is completely formed at a calcination temperature above 550 °C. With increasing calcination temperature, the characteristic diffraction peaks become sharper, narrower, and more intensive, indicating an increase in the crystallinity and particle size of ZnAl_2_O_4_. 

In addition, the average crystallite size (D) of single-phase spinel samples can be estimated by using the Debye–Scherrer equation [21,22]:$$D=\frac{0.9\lambda }{\beta \mathrm{Cos}\theta },$$
where D represents the average crystallite size, λ denotes the wavelength of the X-ray source (Cu Kα, 1.54 Å), β denotes the integral breadth of the (311) diffraction peak, and θ denotes the Bragg’s diffraction angle. The calculated results are presented in Table 1. In normally ordered spinel structures, the intensity of all odd reflections (e.g., 331) shows higher intensity than those of all even reflections (e.g., 400). The XRD peak intensities of (400) and (331) for various calcination temperatures are shown in Appendix A. It was observed that the peak intensities of (400) and (331) changed significantly with increasing calcination temperature. The degree of the order parameter in spinel samples can be calculated by using the following equation [23]:
$$\Delta I=\frac{I_{o}}{I_{o}+I_{e}},$$
where I_o_ and I_e_ denote intensities corresponding to odd and even reflections, respectively. The disordered spinel phase can be formed because some fraction of Al^3+^ cations occupies the tetrahedral site, and Zn^2+^ cations occupy the octahedral site [24,25]. The degree of the order parameter in the spinel samples is shown in Table 1. The order parameters of ZnAl_2_O_4_-550 and ZnAl_2_O_4_-650 are 0.4784 and 0.4922, respectively. These values prove that the ZnAl_2_O_4_ spinel structure obtained at the calcination temperatures of 550 °C and 650 °C is a disordered structure, where Al^3+^ exists in tetrahedral positions and Zn^2+^ in octahedral positions. When the calcination temperature is raised above 750 °C, the degree of the order parameters is above 0.5, indicating that the ZnAl_2_O_4_ spinel is a normally ordered structure.

The FT-IR spectra of the fresh catalysts are shown in Figure 3A. The two vibration bands at 661 and 554 cm^−1^ can be attributed to the stretching mode of the octahedrally coordinated Al-O (AlO_6_), with another assignment of the peak at 495 cm^−1^ to the bending mode of AlO_6_. A broad shoulder band from 704 to 900 cm^−1^ can be assigned to tetrahedrally coordinated Al-O (AlO_4_) [16,26,27,28]. The enlarged spectra (from 600 to 1000 cm^−1^) focusing on the vibration bands of AlO_4_ and AlO_6_ are depicted in Figure 3B. The relative intensities of AlO_4_ to AlO_6_ decrease as the calcination temperature increases. The AlO_4_ vibration bands for the ZnAl_2_O_4_-750 and ZnAl_2_O_4_-850 catalysts are weak. In contrast, the shoulder of the AlO_4_ vibration for ZnAl_2_O_4_-550 and ZnAl_2_O_4_-650 is strong and observable. The presence of Al^3+^ in the tetrahedral position reflects a partial inversion of the ZnAl_2_O_4_ spinel structure. The FT-IR results are consistent with the XRD results and the inversion parameter obtained from the XPS measurement, which is shown later.

The deconvoluted XPS results of the fresh ZnAl_2_O_4_-550 and ZnAl_2_O_4_-650 catalysts are depicted in Figure 4. The Al2p XPS spectrum is illustrated in Figure 4A. The peak at a binding energy of 71.0 eV can be assigned to Al^3+^ occupying the tetrahedral sites (AlO_4_) [29], and a peak of approximately 74.2 eV can be assigned to Al^3+^ occupying the octahedral sites (AlO_6_) [29,30,31]. In Zn2p_3/2_ spectra, as shown in Figure 4B, the larger peak at a lower binding energy (approximately 1022–1022.2 eV) is attributed to Zn^2+^ occupying the tetrahedra sites (ZnO_4_), and a smaller peak at a higher binding energy (approximately 1024.5 eV) is attributed to Zn^2+^ occupying the octahedra sites (ZnO_6_) [29,30,32]. The normally ordered ZnAl_2_O_4_ spinel structure contains Al^3+^ cations at the octahedral sites and Zn^2+^ cations at the tetrahedral sites. When some Al^3+^ cations substitute Zn^2+^ cations in the tetrahedral positions and some Zn^2+^ cations substitute Al^3+^ cations in the octahedral positions, this results in the formation of a partially inversed spinel structure. The inversion parameter of a spinel structure can be defined by surface XPS measurements [30]. Table 2 shows the inversion parameter of the fresh ZnAl_2_O_4_-550 and ZnAl_2_O_4_-650 catalysts by calculating the ratio of AlO_4_/(AlO_4_ + AlO_6_) and ZnO_6_/(ZnO_4_ + ZnO_6_), and the inversion parameter values follow the order of ZnAl_2_O_4_-550 > ZnAl_2_O_4_-650. These results help us demonstrate that the degree of disorder of the spinel ZnAl_2_O_4_ structure decreases with increasing calcination temperature. The deconvoluted patterns of O1s can be fitted to three peaks. The peak O_a_ at the highest binding energy (approximately 543.5–534.8 eV) can be assigned to the oxygen weakly bonded with the surface of the catalysts (such as adsorbed H_2_O and O_2_ from the atmosphere) [16,33,34]. The peak at approximately 533.4–533.6 eV (O_b_) can be assigned to the oxygen-deficient regions or oxygen vacancy (O_v_) [33,34]. Finally, the peak at the lowest binding energy (approximately 531.4–531.7 eV) can be assigned to the lattice O in the ZnAl_2_O_4_ phase [16]. In a partially inversed spinel structure or a disordered structure, the substitution of a Zn^2+^ cation with an Al^3+^ cation to generate a Zn^2+^ octahedral site and Al^3+^ tetrahedral site is the main reason for the oxygen vacancy formation on the catalyst surface. An oxygen vacancy was formed to balance the positive charge of the cation caused by the substitution of Zn^2+^ for Al^3+^ at an octahedral site. The intensities of oxygen vacancy follow the order of ZnAl_2_O_4_-550 > ZnAl_2_O_4_-650, which are listed in Table 2.

The NH_3_-TPD measurements of the fresh catalysts are shown in Appendix A, and the acidic sites are summarized in Table 2. The number of acidic sites decreases in the order of ZnAl_2_O_4_-550 > ZnAl_2_O_4_-650 > ZnAl_2_O_4_-750 > ZnAl_2_O_4_-850. The relationship between the calcination temperature and the total number of acidic sites is shown in Figure 5. In this study, the presence of AlO_4_ and ZnO_6_ in the ZnAl_2_O_4_ spinel structure indicates the partially disordered structure of the catalyst. Furthermore, the partially disordered structure ZnAl_2_O_4_ spinel is capable of producing surface acidity on catalysts, which is related to the XPS intensity of O_a_ (in Figure 4) [16]. The high-intensity peak of O_a_ in the ZnAl_2_O_4_-550 catalyst and its dependency on the calcination temperature follow the same trend as the number of acidic sites. 

The relationship between the number of acidic sites of the fresh catalysts and the inversion parameter (intensity ratio of AlO_4_/(AlO_4_ + AlO_6_)), the XPS intensity of oxygen vacancies (O_v_), and the inversion parameter (intensity ratio of ZnO_6_/(ZnO_4_ + ZnO_6_)) are shown in Figure 6. In the normally ordered ZnAl_2_O_4_ spinel structure, Al^3+^ cations occupy the octahedral sites (AlO_6_) with low acidity. Meanwhile, Al^3+^ cations occupying tetrahedral sites (AlO_4_) in the partially inversed ZnAl_2_O_4_ spinel structure can result in higher surface acidity [16,27,35,36,37,38]. The higher Lewis acidity of AlO_4_ than AlO_6_ can be explained by the lower energy acceptor orbital of the AlO_4_ sites. When AlO_4_ contacts the surface, the removal of one of the four oxygen atoms results in the formation of three-coordinated Al with higher Lewis acidity strength [39]. Figure 6 depicts the positive relationship between the inversion parameter (AlO_4_/(AlO_4_ + AlO_6_)) and total surface acidity. With an increase in the calcination temperature, the disordered ZnAl_2_O_4_ structure decreases, indicating that there are fewer sites in which Al^3+^ cations occupy the tetrahedral position (AlO_4_), which decreases the total surface acidity. Another factor increasing surface acidity is the Lewis acidity of surface oxygen vacancies. Substitution of Zn^2+^ cations for Al^3+^ cations at the octahedral site and the formation of oxygen vacancies (O_v_) to balance the positive charge of the cations can increase surface acidity. Mefford et al. [40] reported that the formation of surface hydroxyl groups was related to the Lewis acidity of surface oxygen vacancies, and evidence was provided from XPS O_a_ peaks of H_2_O or O_2_ adsorbed from the atmosphere. As the number of ZnO_6_ sites decreases with increasing calcination temperature, fewer oxygen vacancies are formed, lowering the surface acidity of the ZnAl_2_O_4_-650 catalyst compared with the ZnAl_2_O_4_-550 catalyst. From the above findings, we conclude that at the calcination temperature of 550 °C, a partially disordered ZnAl_2_O_4_ spinel structure is formed, in which some Al^3+^ cations occupy tetrahedral positions, and Zn^2+^ occupy octahedral positions, resulting in the formation of oxygen vacancies, which increases the surface acidity of the catalyst. With increasing calcination temperature, the structure of ZnAl_2_O_4_ gradually returns to normal, and the number of AlO_4_ and ZnO_6_ sites and oxygen vacancies decreases, lowering the surface acidity of the catalyst.

### 3.2. Catalytic Activity

The reaction results (yield, conversion, and selectivity) calculated from gas chromatograms (a typical gas chromatogram is shown in Appendix A) for the glycerolysis of urea over the ZnAl_2_O_4_-550 and ZnAl_2_O_4_-650 catalysts are summarized in Table 3 and Figure 7A. During the reaction, besides the formation of the main product GC, there are also byproducts such as chemicals (2), (3), and (5). After the 5 h experiment, no other by-products are collected, indicating that the carbon balance between reagents and products is close to 100% (Appendix A, see Appendix A). The relationship between the number of acidic sites vs. GC yield and the number of basic sites (Appendix A) vs. byproduct (2) selectivity for each catalyst is depicted in Figure 7B. At a reaction time of 5 h, the ZnAl_2_O_4_-550 catalyst exhibited higher GC yield and lower (2) selectivity (GC yield = 46%, (2) selectivity = 34%) than the ZnAl_2_O_4_-650 catalyst with GC yield = 41% and (2) selectivity = 41%. Previous studies have reported the existence of the intermediate isocyanate (NCO) complex of Zn, which is an active site for the reaction [18,41]. Moreover, it has been proven that the Lewis acidic site acts as an active site for the ammonia (NH_3_) group of Zn NCO complex adsorption [16]. The ZnAl_2_O_4_-550 catalyst has a higher GC yield and more surface acidic sites than the ZnAl_2_O_4_-650 catalyst. 

The formation of the Zn NCO complex on the solid phase can be confirmed by the FT-IR and TGA of the spent catalysts. The FT-IR spectra of the spent catalysts are shown in Figure 8. Besides the bands for AlO_4_ and AlO_6_ being detected at 704 and 661 cm^−1^, respectively, the vibration band of NCO in the Zn NCO complex on the solid phase is detected at 2350 cm^−1^ [42]. There is no information about the NCO band in the fresh catalysts. However, after a 5 h reaction, this band can be detected by FT-IR. The intensity of the NCO peak for ZnAl_2_O_4_-550 is higher than that for ZnAl_2_O_4_-650 because of the higher number of acidic sites in this catalyst. The same trend is presented by TGA and DTGA profiles in Figure 9. The DTGA profiles of the spent catalysts (Figure 9B) show the decomposition of the Zn NCO complex at 273 °C for ZnAl_2_O_4_-550 and 254 °C for ZnAl_2_O_4_-650 [16]. The higher decomposition temperature for the ZnAl_2_O_4_-550 catalyst can be attributed to a stronger crystalline Zn NCO complex with higher thermal stability. The DTGA results are consistent with the FT-IR results. The higher intensity of the Zn NCO decomposition peak in ZnAl_2_O_4_-550 demonstrated that the ability to adsorb the ammonia group in the Zn NCO of AlO_4_ (Lewis acidic site) is more significant in the ZnAl_2_O_4_-550 catalyst than in the ZnAl_2_O_4_-650 catalyst.

Another notable finding is the relationship between surface basicity and byproduct (2) selectivity. The basic site values are shown in Table 3 and Figure 7B, showing that the relationship between the number of basic sites and (2) selectivity is inversely proportional. That is, if the number of basic sites decreases, more byproducts (2) are formed, and the GC yield reduces. Fernandes et al. [43] explained that surface basicity acted as an active site for the (2) byproduct decomposition process. This study shows the same trend: when using ZnAl_2_O_4_-550 with more basic sites for the reaction, (2) selectivity (34%) is lower, whereas it is higher with ZnAl_2_O_4_-650 (41%).

Catalytic reaction routes over a disordered ZnAl_2_O_4_ spinel structure are depicted in Figure 1. As mentioned above, there are several disordered sites over a partially inverted ZnAl_2_O_4_ spinel: AlO_4_ (symbolized as Al*), ZnO_6_ (symbolized as Zn*) and oxygen vacancies (symbolized as O_v_). ZnAl_2_O_4_ catalysis follows the heterogeneous reaction pathways. The catalysts form the Zn NCO complexes via the reaction with urea. The AlO_4_ site works as a Lewis acidic site where the ammonia (NH_3_) group of the Zn NCO complex can adsorb. The presence of the Zn NCO complex adsorbed on the solid phase is confirmed by the FT-IR and TGA measurements.

In Table 4, the catalytic reaction results in this work are compared to those recently published in the literature. Because the ZnAl_2_O_4_ catalyst follows only the heterogeneous catalysis pathway, the glycerol conversion and GC yield values are relatively low. However, it can still be compared with some previous studies. Zhang et al. prepared the zinc glycerolate (ZMG) and ZnO from zinc glycerolate (ZnO from ZMG) via the calcination method. Glycerol conversion and GC yield of the ZMG are about 65 and 55%, respectively. The authors have demonstrated through reaction tests that catalysts with acidic and basic properties favor the synthesis of glycerol carbonate.

## 4. Conclusions

In this study, the disordered ZnAl_2_O_4_ catalysts were successfully prepared by using the citrate complex method at different temperatures ranging from 550 °C to 850 °C. The ZnAl_2_O_4_-550 catalyst exhibited a more disordered ZnAl_2_O_4_ structure than the other catalysts calcined at a higher calcination temperature, resulting in more disordered sites of AlO_4_ and more oxygen vacancies in ZnAl_2_O_4_-550. The ZnAl_2_O_4_-650 catalyst having Al^3+^ cations at the octahedral sites (AlO_6_) exhibited low acidity, whereas the AlO_4_ sites in ZnAl_2_O_4_-550 resulted in high surface acidity. From the XPS intensities of O_a_ (H_2_O or O_2_ adsorbed) and O_b_ (oxygen vacancies), ZnAl_2_O_4_-550 contained more surface oxygen vacancies than ZnAl_2_O_4_-650, which contributed to the increased surface acidity of ZnAl_2_O_4_-550. The high surface acidity of ZnAl_2_O_4_-550 resulted in strong interactions with the Zn NCO complex on its surface, improving catalytic performance. The intermediate product of urea glycerolysis was 2,3-dihydroxypropyl carbamate (2), which had higher selectivity for ZnAl_2_O_4_-650 than ZnAl_2_O_4_-550, lowering the GC yield. 

## Data Availability

The data presented in this study are available on request from the corresponding author.

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
