# Peer review of "Effect of Calcination Temperatures on Surface Properties of Spinel ZnAl2O4 Prepared via the Polymeric Citrate Complex Method—Catalytic Performance in Glycerolysis of Urea"

_nanomaterials, 2023, doi:10.3390/nano13131901_

Round 1
Reviewer 1 Report
This paper clarifies in detail the relationships between catalytic activity, products selectivity, Zn and Al ions substitution in the tetrahedral and octahedral positions, surface acid and oxygen vacancy. The article is well written and has a high academic level. It should be published after the modification is made as below.
1. In order to make it easier for the readers to understand the test steps, the author should give a schematic diagram of the catalyst activity test device, and the gas chromatographic analysis conditions (such as column temperature, carrier gas type and flow rate, etc.)
2. For sentence of “When some Al3+ cations substitute Zn2+ cations in the tetrahedral positions and some Zn2+ cations substitute Al3+ cations in the tetrahedral positions”, is it right? Maybe, it should be revised as “When some Al3+ cations substitute Zn2+ cations in the tetrahedral positions and some Zn2+ cations substitute Al3+ cations in the octahedra positions
Reviewer 2 Report
The work submitted by Pham et al. is a really interesting and valuable study of influence of calcination temperature on surface properties of disordered ZnAl2O4 nanostructure prepared by polymeric citrate complex method and its catalytic performance in glycerolysis of urea. Authors shown that different calcination temperatures affected on the surface properties of the catalysts, including oxygen vacancies. The ZnAl2O4-550 shows the best catalytic performance which authors connected with a stronger interaction of the Zn NCO complex on its surface. The synthesized catalysts were characterized by fruitful combination of different techniques. It should be mentioned that the manuscript is well-written, the introduction clearly lays out reasons for studying of such system. The work has been carefully done and the results sound. However, I have quite some comments listed below, which have to be addressed before the manuscript can be accepted.
- Authors claimed that the calcination @ 750 and 850°C leads to the formation of ZnAl2O4 spinel is a normally ordered structure (based on the XRD data). From the XPS data obtained for the catalyst calcined @ 550 and 650°C authors calculated inversion parameters based on the ratio between the Al3+ and Zn2+ located in different crystallographic positions. From my opinion the Al2p and Zn2p3/2 XP spectra measured for the catalyst calcined @ 750 and 850°C should be presented as well. Is it really only single species presented in it?
- Concerning the calculation of intensity of Oxygen Vacancies (Table 2). How it was done? As I understand it is just intensity of O1s state assigned to the Oxygen Vacancy, marked Ob. I do not think that this is legal to use pristine intensity for the comparison between the samples, typically for the XPS intensity should be normalized some how on the «standard». For example it could be the fraction of Ob, contributing to the total O1s intensity, which is: Fraction (Ob), % = I(Ob)*100/I(Ototal).
- Concerning the O1s deconvolution (Figure 4) - as I can see the parameters of peaks fitting for ZnAl2O4-550 and ZnAl2O4-650 are different for the same species (fwhm and G/L) – I do recommend to the authors make the fitting with the same parameters for the same species and calculate the Ob fractions as I mentioned above.
- Also I recommend starting the description of XPS results from the XP spectra and then describe the quantitative data obtained from peak fitting.
- Table 1 and Table 2 for each parameter the method from which it was obtained has to be mentioned in the table.
Reviewer 3 Report
Manuscript Number: nanomaterials-2423503
Full Title: Effect of calcination temperatures on surface properties of disordered ZnAl2O4 nanostructure prepared by polymeric cit-3 rate complex method—catalytic performance in glycerolysis of urea
Remarks to the Authors:
In this study, the authors investigated the process of urea glycerolysis using ZnAl2O4 catalysts that were prepared using a citrate complex method. The focus of the research was to understand the influence of calcination temperatures on the surface properties of the catalysts. The authors varied the calcination temperature in the range of 550°C to 850°C and examined its impact on the catalysts. The reciprocal substitution between Al3+ and Zn2+ cations during the preparation of ZnAl2O4 led to the formation of a disordered bulk ZnAl2O4 phase. The authors observed that different calcination temperatures strongly affected the surface properties of the ZnAl2O4 catalysts, particularly the presence of oxygen vacancies. Among the various catalysts prepared, the ZnAl2O4-550 catalyst exhibited a large specific surface area and highly disordered surface sites. This resulted in increased surface acidity, which enhanced the interaction of the Zn NCO complex on the catalyst surface, ultimately leading to improved catalytic performance. To analyze the catalyst performance and reaction products, the authors employed FTIR and TGA on the spent catalysts. The results revealed that the ZnAl2O4-550 catalyst exhibited a greater formation of a solid Zn NCO complex compared to the ZnAl2O4-650 catalyst. Consequently, the ZnAl2O4-550 catalyst demonstrated superior performance in terms of glycerol conversion (72%), glycerol carbonate yield (33%), and reduced formation of byproducts. Overall, I can recommend this manuscript for publication in MDPI Nanomaterials but only after major revision. Please find my comments below.
1. First, the authors should present GC-FID or GC-MS spectra with obtained product mixture. This information can be included in the Support Information file. Also, the authors should decrypt/label each intense/main peak observed on spectra.
2. What is the experimental error of the catalytic activity study (selectivity, conversion, yield)? Did the authors check the reproducibility of the obtained experimental data? This information must be provided.
3. Did the authors calculate the carbon balance? I think this information must be presented.
4. It is not clear why the title, abstract, and support information file are partly highlighted in yellow. What does it mean?
5. The authors should provide the real composition of synthesized catalysts and include it in Table 1. ICP-OES or XRF analyses can be useful.
6. The authors must provide the original spectra for NH3-TPD and CO2-TPD analyses.
7. The authors should compare their achieved results (conversion, selectivity, yield, etc.) with well-known literature data (recently published) and present it in a table format.
8. The authors should better emphasize/disclose the novelty of their study in the Abstract and Conclusions sections.
9. Unfortunately, I can not find any discussion related to the reaction mechanism. I think the reaction pathway with appropriate discussion should be presented.
Minor editing of English language required
Round 2
Reviewer 2 Report
The authors improved the manuscript, currently it is suitable for the journal and could be accepted in a present form.
Author Response
The authors improved the manuscript, currently it is suitable for the journal and could be accepted in a present form.
Response) We appreciate this kind comment.
Reviewer 3 Report
The authors significantly improved their manuscript, but I still have one question regarding the calculation of the carbon balance (question 3) in Table S1. I don't understand how the authors achieved an almost 100% carbon balance when the initial amount of glycerol and urea is more than 400 mmol, while the amount of products and glycerol after the reaction is slightly over 200 mmol. The authors should provide an explanation for this discrepancy. It's possible that they forgot to include the amount of urea after the reaction, so they need to address this issue and clarify it.
